# Speech Discrimination in Infancy Predicts Language Outcomes at 30 Months for Both Children with Normal Hearing and Those with Hearing Differences

**DOI:** 10.3390/jcm11195821

**Published:** 2022-09-30

**Authors:** Kristin M. Uhler, Sean R. Anderson, Christine Yoshinaga-Itano, Kerry A. Walker, Sharon Hunter

**Affiliations:** 1Department of Physical Medicine and Rehabilitation, University of Colorado School of Medicine, Children’s Hospital Colorado, Aurora, CO 80045, USA; 2Department of Physiology and Biophysics, University of Colorado School of Medicine, Aurora, CO 80045, USA; 3Institute of Cognitive Science, University of Colorado, Boulder, CO 80309, USA; 4Department of Otolaryngology—Head & Neck Surgery, University of Colorado School of Medicine, Aurora, CO 80045, USA; 5Department of Psychiatry, University of Colorado School of Medicine, Aurora, CO 80045, USA

**Keywords:** infant speech discrimination, early language, hearing aids, hearing loss

## Abstract

Background: Speech discrimination assessments are used to validate amplification fittings of older children who are hard of hearing (CHH). Unfortunately, speech discrimination is not assessed clinically ≤24 months and in turn no studies have investigated the relationship between speech discrimination during infancy and later language development among CHH. Objective: To examine the relationship between an individual infant’s speech discrimination measured at 9 months and their expressive/receptive spoken language at 30 months for children with normal hearing (CNH) and CHH. Methods: Behavioral speech discrimination was assessed at 9 months and language assessments were conducted at 16, 24, and 30 months using a parent questionnaire, and at 30 months using the Mullen Scales of Early Learning among 90 infants (49 CNH; 41 CHH). Results: Conditioned Head Turn (CHT) performance for /a-i/ significantly predicted expressive and receptive language at 30 months across both groups. Parental questionnaires were also predictive of later language ability. No significant differences in speech discrimination or language outcomes between CNH and CHH were found. Conclusions: This is the first study to document a positive relationship between infant speech discrimination and later language abilities in both early-identified CHH and CNH.

## 1. Introduction

Following the National Institute of Health Consensus Development Conference on Early Identification of Hearing Impairment in Infants and Young Children [1] in 1993 and the JCIH 1994 Position Statement [2], universal newborn hearing screening (UNHS) programs began to be established in the United States, such as those in Colorado [3], Rhode Island [4] and Hawaii [5]. A recently published systematic review found that UNHS/EHDI (Early Hearing Detection and Intervention) programs resulted in earlier identification, earlier intervention, and better developmental outcomes [6,7]. As a direct result of UNHS/EHDI programs, more children are confirmed as deaf or hard of hearing (CHH) by three months of age, have been fitted with amplification technology by four months, and enrolled in early intervention services in the first few months of life. UNHS/EHDI programs strive to meet EHDI 1-3-6 goals (screen by one month, identify by three months, and enroll in early intervention services by six months) [8]. 

Despite these significant advances in early identification and intervention, procedures to assess speech discrimination in very young infants and children before the acquisition of spoken language remain limited. Recent findings among CHH between 6 and 17 months of age who benefited from UNHS/EHDI suggest that vowel sounds (e.g., /a-i/) are more likely to be discriminated than consonant sounds (e.g., /ba-da/) [9,10,11,12,13]. However, assessing speech discrimination, aside from parent questionnaires, is quite limited until after two years of age and, more commonly, after three years of age [14,15,16]. Without speech discrimination assessments, validation of amplification fitting cannot be done. Modifications and development of early intervention strategies for listening and spoken language will also be limited without additional information about a child’s prelinguistic speech discrimination skills. No studies investigating the relationship between speech discrimination measured during infancy and language outcomes among early-identified CHH were found in the literature. Until recently, assessment of speech discrimination during infancy and toddlerhood has yielded only group results, limiting its utility for individuals [9,10].

Research among children with normal hearing (CNH) suggests that speech discrimination skills at six months of age assessed through Conditioned Head Turn (CHT) techniques [17] predict MacArthur Communicative Development Inventory scores at 24 months, suggesting that speech discrimination in infancy plays a significant role in spoken language acquisition. Although CHT techniques have been used to study speech discrimination in CNH, no studies have been reported for CHH below the age of 3 years, either comparing their performance to children with typical hearing or investigating the relationship of their early speech discrimination with later language development. This study aims to examine the relationship between an individual infant’s speech discrimination measured at nine months of age and an infant’s expressive and receptive spoken language measured at 30 months of age. 

Since the inception of UNHS/EHDI, research has revealed a wide array of demographic characteristics, such as a lesser degree of hearing loss [18,19,20] and earlier age at amplification [6,7,18] which are related to improved language outcomes and cognitive abilities [7,18,20,21]. Ching et al. [22,23] reported that maternal level of education, degree of hearing loss, gender, and presence of additional disabilities were significant predictors of language outcomes. While understanding the relationships between these characteristics and language outcomes is essential, variables such as gender, maternal level of education, and degree of hearing loss cannot be altered to impact outcomes. However, performance on speech discrimination can be used to guide interventions; for example, the assessment of speech discrimination in older children [24,25] and adults [26,27] with hearing differences validates that the amplification fitting results in access to the sounds of their spoken language, and is related to communication. Performance on speech discrimination guides interventions such as hearing aid programming.

Hearing aids are programmed using a computer, an individual’s hearing thresholds, and proprietary software from the hearing aid manufacturer. The resulting gain of the hearing aids is verified by measuring the output in the individual’s ear canal using specialized equipment (i.e., Audioscan Verifit, Dorchester, Ontario). The gain/output of the hearing aid is compared to the prescription for each listener based on age, amount of hearing loss, and the prescription being used (i.e., Desired Sensation Level; DSL) [28]. How well hearing aids meet the prescriptive gain targets is influenced by changes in audiometric thresholds and ear canal size, which change as children grow [29,30,31]. Among older CHH, hearing aid characteristics, such as how well they are programmed to meet the prescriptive hearing aid fitting target, impact the audibility of speech sounds of the wearer and are related to better language outcomes [18,21]. Even among CHH who used hearing aids and have good aided audibility, speech discrimination abilities first assessed at three years of age remained highly variable [20]. These results reinforce that verifying the output in the infant’s ear canal is insufficient to ensure optimal speech discrimination abilities. Unfortunately, speech discrimination assessment before two years is not included in audiologic diagnostic evaluations for young infants and toddlers. Therefore, in an infant population it is not known whether infants can discriminate speech sounds during a period of rapid language learning, under what conditions they can discriminate speech sounds (e.g., quiet, noise), and what the variability in speech discrimination performance is among CHH or CNH. 

We propose that the earlier infants/toddlers are fit well with amplification, the more likely that they will be able to discriminate speech sounds early in life and then achieve higher language outcomes. Therefore, the ability to assess speech discrimination in the first year of life and its relationship with language abilities assessed later in childhood is needed. This experiment aimed to assess the relationship between audibility and speech discrimination in infancy and later language abilities. In this manuscript, we report on the effects of audibility and speech discrimination on receptive and expressive spoken language for infants with hearing loss, who met the EHDI benchmarks, and a comparison group of peers with normal hearing. Specifically, this study seeks to demonstrate that well-fit hearing aids (measured by Speech Intelligibility Index; SII) and behavioral speech discrimination scores utilizing a conditioned head turn technique (CHT/Visual Reinforcement Infant Speech Discrimination; VRISD) in the first year of life predict spoken language outcomes measured at 30 months of age. Additionally, we assess the relationship between a commonly used parent questionnaire, the MacArthur-Bates Communicative Development Inventories, (MBCDI; [32]), and an in-clinic administered test, The Mullen Scale of Early Learning (MSEL; [33]). The following questions were examined:Is there a difference in speech discrimination abilities between CHH and CNH on /a-i/ and /ba-da/ discrimination?How do early speech discrimination abilities relate to later spoken receptive and expressive language abilities in CHH?At 30 months of age, will scores from a parent questionnaire of their child’s spoken language inventory significantly correlate with their assessed early receptive and expressive spoken language ability?Among CHH, what effect does audibility and hearing aid use have on early speech discrimination and spoken language abilities?

## 2. Materials and Methods

### 2.1. Participants 

Data from 90 infants participating in an ongoing longitudinal study were analyzed. All participants’ hearing was screened via universal newborn screening, and results per parent report were recorded. Participants included 41 CHH (21(M), 20(F)) and 49 CNH (23(M), 26(F)). All were born full-term and healthy (see Table 1). CHH infants were approximately one month older at CHT testing, but other assessments occurred at similar ages. All CHH were enrolled in early intervention and fit with hearing aids by six months of age (see Table 2). Data from 18 CHH and 19 CNH were excluded in the current study because they did not have both CHT and MSEL data for the following reasons: COVID shut downs (6 CHH, 3 CNH), diagnosis with secondary disability (1 CHH), lost to follow-up (4 CHH, 12 CNH), primary spoken language in the home was not English (2 CHH), family relocated (4 CHH, 3 CNH), different testing protocol was used (1 CNH), could not complete conditioned head turn (1 CHH). CHT results for 40 CHH and CNH were previously reported. [12] Maternal level of education was gathered as a demographic variable [34] and categorized into four levels: (1) children whose mothers had a high school diploma or lower (20.0%), (2) children whose mothers reported post-secondary attendance (12.2%), (3) mothers who graduated from college (36.7%), and (4) postgraduate levels of education (31.1%). Demographic information for participants from this study is included in Table 1.

Criteria for inclusion into the study (between 1 and 5 months of age) were (a) no evidence of significant developmental delays or secondary disabilities per parent report, or as indicated in the electronic medical record, (b) absence of fluid in the middle ear day of testing, (c) English as the primary language is spoken in the home, and (d) demonstrated conditioned head turn in visual reinforcement audiometry after 6 months of age.

#### Participant Hearing Aids 

All CHH participants used their personal hearing aids or cochlear implants (two children transitioned to cochlear implants during this study) during the CHT and MSEL testing. Children were fit following diagnosis of hearing loss with bilateral, behind-the-ear hearing aids coupled to custom earmolds, filtered ear hooks, and programmed using DSL v5.0 [28]. Children received individualized care from their managing audiologist following best practices for fitting guidelines, verification, and validation [35], and recommended guidelines [36].

Hearing aid use was collected by reading the average daily data logging values from the child’s hearing aid at the time of CHT and MSEL. The mean data logging recorded was 6.39 hours per day (SD = 4.16) for CHT and increased at MSEL to 8.26 averaged hours per day (SD = 4.70), see Table 2 for characteristics unique to CHH. While hearing aid usage increased an average of two hours per day between CHT testing (nine months) and MSEL assessment (30 months), the increase was not significant, paired t-test (t(28) = −1.827, *p* = 0.079).

### 2.2. Procedures and Materials 

The local institutional Review Board approved this project. Consent was obtained from parents/guardians before beginning the research project. Parents were provided with compensation for their child’s participation. Participant compensation was paid in cash or gift cards at a rate of USD15 per hour for in-person or telepractice visits and USD20 for MBCDI completion.

#### 2.2.1. Stimuli 

The four stimuli used for the experiment were /ba/, /da/, /a/, and /i/. The stimuli were natural speech tokens produced by a female speaker, and adult listeners in the laboratory verified that the stimuli sounded natural. See Figure 1 for the distribution of F1 and F2 for the stimuli used for the CHT paradigm. Stimuli were digitized using a 16-bit analog-to-digital converter (AD Instruments Power Laboratory/16 SP) at 40 kHz and edited using Goldwave, Inc. (St. John’s, NL, Canada). The stimuli were down-sampled to 22,050 Hz and edited to 500 msec duration. The digitized speech stimuli described above were routed to an audiometer for presentation in the sound field. During testing, stimuli were presented at 1200 msec interstimulus intervals. Stimuli were loudness equalized and presented at either 50, 60, and/or 70 dB SPL-A. See Uhler et al. [10] for additional methodological details.

#### 2.2.2. Speech Discrimination

Two sessions were required to complete the speech discrimination protocol. The first session consisted of the case history (information related to the infant’s general health, development, and years of education of the infant’s mother), screening for middle ear fluid (i.e., tympanometry), hearing screening, and, if time allowed, a threshold search for /a/ using CHT. The second visit consisted of the threshold search for /a/, if not completed at the first visit, and the CHT assessment protocol. 

Two speech contrasts were assessed (/ba-da/ and /a-i/) during CHT testing. Presentation order was randomized and within each stimulus pair either contrast could serve as the background stimulus. The other speech sound served as the target. The member of the pair serving as the target stimulus was counterbalanced across participants. The infant learned to respond by turning his/her head when the target stimulus was presented. 

Testing was completed in a sound booth. Caretakers accompanied infants into the sound booth for the CHT assessment. The background stimulus was on and being repeated with a 1200 ms interstimulus interval when they entered the room. Infants were either seated on the caretaker’s lap or in a highchair in the center of the room to minimize distractions or task fatigue. Regardless of positioning, the distance between the infant’s head and the speaker was the same. The speaker and visual reinforcement video screen were 90º to the right of the infant’s midline. An assistant who centered the infant’s gaze was positioned in front of the infant, slightly to their left. During testing, the caretaker and the assistant listened to music through supra-aural headphones to prevent them from hearing the sounds presented and inadvertently reinforcing the child or alerting the child to a contrast stimulus. The evaluator observed the infant through a window in a separate room outside the sound booth. 

The discrimination task consisted of two phases: conditioning and testing. In the conditioning phase, only change trials were conducted so infants could learn the association between a change in the sound and the reinforcer. During conditioning, the target sound was presented at a louder level than the background sound (+6 dB SPL-A) to alert infants to the sound change. Initially, the reinforcer was automatically activated after two target sounds were presented. After the infant made two consecutive head turns before the end of the first two presentations of the target sounds (i.e., anticipatory head turns), the intensity cue was removed. Once testing began, the evaluator could no longer hear the stimuli. Trials were initiated by a button press once the evaluator determined the child’s attention was directed toward the midline.

Computer software determined trial-type presentation, with either 7 or 8 of the 15 trials being a change or no-change trial as randomly determined by the computer; the evaluator was blind to trial type. If the trial was a no-change trial, the background sound was presented three times. If the trial was a change trial, the target sound was presented three times. At the end of each trial, the background sound continued. The evaluator indicated when the child executed a head turn toward the speaker by button press. The CHT software determined if the child’s head turn was a correct response to a change trial or a false positive to a no-change trial. Correct responses were rewarded by the automatic presentation of a visual reinforcer, an animated video. Performance on the task was quantified using d-prime [37]. d’ is calculated by using the z-score:
d’ = inverse norm((false alarms)/(correct rejections + false alarms)) − inverse norm ((hits)/(hits + misses)).(1)

The advantage of d’ is that it eliminates the effect of response bias, as the calculation considers the number of false positives/alarms and hit rate. A “false alarm” occurs when a child turned their head, but no change occurs in the stimuli (/a-a-a/). In this case, the button indicating that a head turn occurred would be pressed, but no reinforcement would occur. In contrast, a “hit” occurs when a child turns their head in response to a change in stimuli, which would lead to the behavior being rewarded. If the child achieved a d’ of at least 1.2 at 50 dB SPL-A, then testing was complete [10]. The rationale was that a child able to discriminate at a low presentation level would also be able to discriminate at higher levels [38].

If the child did not reach criterion at 50 dB SPL-A, the level was increased to 70 dB SPL-A, and testing resumed. Once 15 trials at 70 dB SPL-A were completed, the presentation level was reduced to 60 dB SPL-A, and 15 trials were completed at that presentation level, regardless of whether or not a child reached criterion, at 70 dB SPL-A. Therefore, for children who did not reach criterion at 50 dB SPL-A, a total of three conditions (e.g., /a-i/ at 50, 60, and 70 dB SPL-A) were completed. In each session, testing continued until all conditions were completed, or the child was too fussy or tired to continue. On average, a single condition (i.e., /a-i/ at 50 dB) was completed in an average of 5 minutes and 32 seconds (SD = 5.35 min). Because some infants who reached criterion were only tested at 50 dB SPL-A, the score for which criterion was reached was used in the analysis, regardless of intensity. For infants who did not reach criterion, their best score was used in the analysis, regardless of intensity. To determine the expected range of d’ values if children were performing at chance, a Monte Carlo study of 200,000 simulations indicated that d’ values equal to or above 1.28 could be considered significantly above chance (one-sided, 95% confidence interval).

#### 2.2.3. Receptive and Expressive Spoken Language Assessment

When the children turned 30 months of age, their caretakers brought them to the laboratory, or during the COVID-19 pandemic, testing was completed via telepractice (*n* = 4). During this visit, the MSEL [33] was administered by a trained researcher. The MSEL allows assessment of higher-level linguistic abilities compared to a parent questionnaire; for example, the receptive portion of the test requires children to follow single to multi-step instructions or point to pictures based on instructions. The expressive language portion assesses vocabulary, sentence complexity, and the ability to repeat back words or sentences and answer complex questions. The complete MSEL involves the examiner eliciting specific cognitive, motoric, visuoperceptual, and communicative behaviors from the child. For our analyses, we used the raw scores from the receptive and expressive spoken language scales. While standardized scores allow comparison of an individual’s performance to typical performance in a larger reference group [39], they reduce the ability to see smaller differences in performance between individuals and may obscure the relationship with other measures of individual performance (e.g., d’). 

#### 2.2.4. Early Language Inventory

As part of our larger longitudinal study, early spoken language was assessed through the use of the MBCDI Words and Sentences at 16, 24, and 30 months of age [32]. Parents are instructed to mark words that their child uses from a catalog of choices as well as answer questions pertaining to how their child uses words, sentence complexity, and grammar. MBCDI questionnaires for the 24- and 30-month timepoints were anchored to the most previous response filled out by the parent or guardian for convenience. MBCDI parental questionnaires were sent to the family electronically or by mail at each time point. However, for the 30-month MBCDI, if responses were not received by the time of the MSEL assessment, parents completed the questionnaire in person. The MBCDI responses were scored using an online program from mb-cdi.standford.edu (CDI Advisory Board, Stanford, CA, USA). 

### 2.3. Statistical Analysis

The CHT score used for the analysis was the best d’ score regardless of intensity, summarized in Table 3. The significance (or alpha) level was set to 0.05 and all analyses were complete in version 4.2.1 of R [40]. Descriptive statistics were summarized as mean (M) and standard deviations (SD), sample size (*n*), and percent, as appropriate. Effect sizes were calculated using Cohen’s partial f^2^. Multivariate linear regression or ANOVA was used to model each outcome of interest (MSEL receptive and expressive language at 30 months) as a function of CHT score. During analysis, CHH were separated into two groups according to hearing status (mild (16–40 dBHL) or moderate ≥ 41 dBHL). Mixed-effects regression and analysis of variance (ANOVA) were implemented via version 1.1–29 of the *lme4* [41] and version 3.1–3 of the *lmerTest* [42] packages in R in cases where multiple levels of observation within subjects were present (e.g., different presentation levels or multiple timepoints of one measure). In some of these cases, multivariate regression and ANOVA (MANOVA) were implemented following the procedures outlined by Snijders and Bosker [43]. Cohen’s partial f^2^ was calculated using version 0.7.0 of the *effectsize* package in R. Post-hoc t-tests with Tukey corrections for multiple comparisons were completed using version 1.7.5 of the *emmeans* package in R.

## 3. Results

### 3.1. CHT Performance

Results from the CHT as an effect of intensity level are shown in Figure 2 for CNH and CHH and summarized in Table 3. Eighty-one percent of CNH were able to discriminate the /a-i/ contrast and 51% were able to discriminate /ba-da/ for at least one intensity level. For CHH, 76% were able to discriminate the /a-i/ contrast and 58% were able to discriminate /ba-da/ for at least one intensity level. If an infant reached criterion at 50 dB SPL-A, they were not tested at 60- and 70-dB SPL-A, see Table 3. Each infant’s score is only represented at one intensity level, which is the highest d’ value observed. 



**Question 1: Is there a difference in speech discrimination abilities between CHH and CNH on /a-i/ and /ba-da/ discrimination? Does maternal level of education impact speech discrimination?**



A mixed-effects MANOVA with fixed-effects of presentation level (50, 60, or 70 dB SPL-A) and hearing status (CNH (0–15 dB HL), Mild (16–40 dB HL), or ≥Moderate (≥41 dB HL)), as well as a random effect of child, was completed. Only children who completed the CHT task at all levels were included in the analysis. This included 22 CNH, 13 children with mild hearing loss, and 9 children with ≥moderate hearing loss. Maternal level of education and possible interactions with hearing status and presentation level were excluded from the model because a model comparison test revealed that including them did not significantly improve model fit. Results are shown in Table 4. For the /a-i/ contrast, results indicated that there was a significant effect of presentation level of CHT but not hearing status. Post-hoc tests are shown in Table 5 and revealed that d’ was significantly better for 70 dB SPL-A compared to 50 dB SPL-A but not between other presentation levels. Similarly, for the /ba-da/ contrast, there was a significant effect of presentation level but not hearing status. Post-hoc tests did not reveal significant differences between presentation levels, though the difference between 70 dB SPL-A and 50 dB SPL-A fell just above the threshold for significance. When maternal level of education was included in the model, it was not a significant predictor of speech discrimination performance for either /a-i/ or /ba-da/ contrasts. 



**Question 2: How do early speech discrimination abilities relate to later receptive and expressive spoken language abilities?**



### 3.2. MSEL Receptive Spoken Language 

The relationship between the speech discrimination as measured by CHT for the /a-i/ contrast and language outcomes is shown in Figure 3. The mean raw score for receptive spoken language ability was 30.3 (SD = 3.7) for CHH and 30.7 (SD = 3.7) for CNH. The output of a fixed-effects multivariate regression including dependent variables MSEL Receptive and Expressive spoken Language is summarized in Table 6. The regression included fixed-effects of CHT /a-i/ contrast and maternal education level. These effects were included in the model because they resulted in a significantly improved fit relative to a model including only CHT /a-i/ contrast, but no significant improvement for more complicated models including effects of hearing status or possible interactions were found. Among all children, the CHT score was positively associated with MSEL receptive spoken language at 30 months. Figure 3 displays the estimated regression line for receptive spoken language ability as a function of CHT score at the mean maternal level of education. The figure also depicts the 50^th^ percentile of the age-referenced score, allowing readers to have a reference to the standard scores (dotted line). Note that for each 1 unit increase in CHT score (a value between −0.47 and 3.00), there is an estimated 1.06 point increase in MSEL receptive spoken language score (95% CI = 0.0 to 2.1). When hearing status was included in the model, it was not significant, which can be confirmed by the substantial overlap between CNH and CHH in Figure 4. To examine the goodness-of-fit of CHT on receptive spoken language outcomes, two separate linear regressions were fit to CNH and CHH including an interaction term with maternal level of education. The adjusted R^2^ for the CNH and CHH groups were 0.14 and −0.01, respectively, suggesting that goodness-of-fit was poorer for the CHH compared to CNH group. 

### 3.3. MSEL Expressive Spoken Language 

The mean expressive language raw scores were 28.8 (SD = 4.6) for CHH and 29.4 (SD = 4.3) for CNH. Among all children, the CHT score was positively associated with MSEL expressive language at 30 months (see Table 6). Figure 3 displays the estimated regression line for expressive language ability as a function of CHT score, and the 50th percentile of the age-referenced score, allowing readers to have a reference to the standard scores (dotted line). For each 1 unit increase in CHT score, there is an estimated 1.64 point increase in MSEL expressive spoken language score (95% CI = 0.4 to 2.9). Similar to receptive spoken language, expressive spoken language scores overlapped substantially between CHH and CNH. To examine the goodness-of-fit of CHT on expressive spoken language outcomes, two separate linear regressions were fit to CHH and CNH including an interaction term with maternal level of education. The adjusted R^2^ for the CNH and CHH groups were 0.24 and 0.08, respectively, suggesting that the goodness-of-fit was poorer for the CHH compared to CNH group. In contrast to /a-i/, regressions including /ba-da/ contrasts used to predict spoken language outcomes revealed no significant relationship between receptive or expressive spoken language.



**Question 3. At 30 months of age, will scores from a parent questionnaire of their child’s spoken language inventory significantly correlate with their assessed early receptive and expressive spoken language ability?**



### 3.4. MBCDI Words Produced

Results from the MBCDI words produced (reported in percentile) are shown in Figure 5 over 16, 24, and 30 months. A mixed-effects ANOVA was completed including fixed-effects of time (treated as a factor), MSEL receptive raw score, a time × MSEL interaction, and hearing status. This model showed a significant improvement in fit relative to a model including only the effect of time. Results revealed that the percentage of words produced increased over time (see Table 7). Post-hoc tests revealed that there was a significant increase in the MBCDI score between 16 and 30 months and 16 and 24 months but not 24 and 30 months (see Table 8). The MSEL receptive spoken language raw score was also significant. The MBCDI percentile was also predicted by a significant time (categorical) × MSEL receptive spoken language (continuous) interaction. The largest coefficient was at 24 months, suggesting that MBCDI scores at 24 months have the strongest relationship with MSEL receptive language scores. Hearing status did not significantly affect the MBCDI percentile of words produced. 

Results with expressive language were similar. Results revealed that the percentage of words produced increased over time (see Table 7). Post-hoc tests showed a significant increase in the MBCDI score between 16 and 30 months and 16 and 24 months but not 24 and 30 months (see Table 8). The MSEL receptive language raw score was also a significant predictor of the MBCDI percentage of words produced. There was also a significant time × MSEL receptive spoken language interaction. The largest coefficient was at 24 months, suggesting that it might have the strongest relationship with MSEL expressive spoken language scores. Hearing status did not significantly affect MBCDI percentile of words produced. When maternal level of education was included in either model, it did not significantly predict the MBCDI percentile of words produced. In both cases, including a time × MSEL interaction in the model significantly improved the fit, suggesting that the MBCDI responses at 24 months are most strongly related to receptive and expressive spoken vocabulary assessed at 30 months by the MSEL.

The relationship between MBCDI words produced at 30 months and CHT was assessed via regression. Upon initial analysis with multiple linear regression using the same predictors as Table 6 (sensitivity to /a-i/ and maternal level of education), model diagnostics revealed curvilinear residuals and a departure from the assumption of normality. Using a logit regression and including a squared term for sensitivity to /a-i/ resolved the issue. Results revealed no significant relationships between the speech discrimination for the /a-i/ contrast or maternal level of education with MBCDI words produced.



**Question 4: How do receptive and expressive language abilities in CHH compare to CNH?**



Language outcomes are shown in Figure 4 and Table 9. They demonstrate overlap between CHH and CNH on both receptive and expressive language. Results of a multivariate ANOVA including receptive and expressive language raw scores as dependent variables and hearing status as the predictor showed no significant effect of hearing status on language outcomes F(4,174) = 0.575, *p* = 0.681.



**Question 5: Among CHH, what effect does audibility and hearing aid use have on early speech discrimination and language abilities?**



### 3.5. Speech Intelligibility Index and Data Logging among CHH

We examined the relationship between aided SII at two time points, at the CHT visit and at the MSEL visit, which are reported in Table 10. Aided SII was positively related to CHT performance for either the /a-i/ at 50, 60 and 70 dB SPL-A and /ba-da/ contrast for 50 dB SPL-A and 60 dB SPL-A. Aided SII, measured at 9 months (time of CHT) for 50, 60, and 70 dB(A), was not significantly correlated with MSEL receptive or expressive vocabulary raw scores. Similar relationships were observed for the aided SII measured when the child participated in MSEL. Aided SII measured at the time of the MSEL (30 months) at 50, 60, and 70 dB(A) was not significantly correlated with MSEL receptive or expressive language raw scores. Figure 6 reveals very little variability in aided SII values for children with mild hearing loss, but greater variance among CHH with ≥moderate hearing losses. 

Next, we examined the relationship between data logging and speech discrimination and language abilities (see Figure 6 and Table 10). Datalogging (i.e., average amount of daily hearing aid use) measured at the time of CHT (9 months) was not significantly correlated with MSEL receptive or expressive spoken vocabulary raw scores. Datalogging at the time of the Mullen (30 months) was not correlated with MSEL receptive or expressive spoken vocabulary raw scores. While datalogging measured at the time of CHT (9 months) was not correlated with maternal level of education, there was a much stronger significant correlation at the time of MSEL (30 months).

## 4. Discussion

The current study’s primary aim was to examine whether an individual infant’s speech discrimination score at nine months predicted receptive and expressive spoken language abilities at 30 months of age among CHH and CNH. Speech discrimination measured at 9 months of age using the /a-i/ contrast predicted both receptive and expressive spoken language at 30 months. These results demonstrated, for the first time, that a clinically recommended tool, CHT, for assessing each infant’s speech discrimination performance at nine months of age [44] significantly predicts both receptive and expressive spoken language measured by MSEL at 30 months of age. Previous studies have revealed a significant relationship between a parent report measure, the MBCDI, and speech discrimination [17] among CNH. We have found an association between speech discrimination and a direct objective measure, MSEL, of both receptive and expressive spoken language for both CHH and CNH.

### 4.1. Speech Discrimination 

To address our first questions: (1) is there a difference in speech discrimination abilities between CHH and CNH on /a-i/ and /ba-da/ discrimination, and (2) if maternal level of education impacts speech discrimination? We found that most infants (CHH and CNH) discriminated /a-i/ at one of three intensity levels (50-, 60-, or 70-dB SPL-A). There was no significant difference in speech discrimination abilities between CHH and CNH. Discrimination of the /ba-da/ contrast was observed in fewer infants across both groups. Of note, all CHH were identified through newborn hearing screenings, fit with hearing aids, and enrolled in early intervention by six months. Among CHH, there was no difference in speech discrimination abilities as a function of the degree of hearing loss (i.e., mild versus ≥moderate). Though non-significant findings are typically not discussed, our findings showed no significant differences between CHH and CNH groups for discrimination of either /a-i/ or /ba-da/. Moreover, these findings within the CHH group, that no significant differences were found as a function of the degree of hearing loss, are theoretically important. These findings highlight the effectiveness of the early identification/intervention as a result of UNHS/EHDI for these CHH on speech discrimination abilities in infancy. Before UNHS/EHDI systems, it would not have been possible to assess infant speech discrimination among CHH [45] because the average age at identification of mild to moderate hearing loss was delayed until around 2 years. Based on the importance of the first year in the development of speech perception abilities [46,47], we postulate that the development of early speech perception would be negatively affected without early identification/intervention and could set off a cascade of delayed development in spoken language for these children.

While all children demonstrated conditioning to the task before testing began, bringing children back may have resulted in more children being able to discriminate the /ba-da/ contrast. Because we aimed to examine the clinical utility of this CHT procedure, we did not design the study for return visits. The logistics of having parents return within a short timeframe would likely increase subject attrition and be challenging to accommodate in future clinical settings. Additionally, assessing speech discrimination for one contrast took an average of 5 min which is feasible to complete in a clinical audiology setting among young infants. 

### 4.2. Speech Discrimination and Language Abilities

Next, we address our second question, is there a relationship between early speech discrimination abilities and later receptive and expressive spoken language abilities? Yes, during the first year, discrimination of the /a-i/ contrast is related to expressive and receptive language abilities as measured by the MSEL at 30 months of age, but not as measured by the MBCDI among CHH and CNH. Infants who were unable to demonstrate a discrimination response at nine months of age had poorer language on average than those who could discriminate these speech sounds. Our present findings are consistent with other studies demonstrating a relationship between discrimination skills, measured both behaviorally and using evoked potentials, during the first year of life, e.g., [17,46,48,49] for speech sounds and later language abilities measured through parent questionnaires. We were surprised that the same relationship was not observed for the /ba-da/ contrast. Perhaps it is due to overall poorer performance on /ba-da/ or lack of variability in performance on this contrast. However, our study design did not allow us to examine “why” /ba-da/ speech discrimination was not related to later language development. 

While our approach was similar to the study design implemented by Tsao et al. [17], there were differences. Previous research [50,51,52] suggests that infants require a higher intensity level to achieve criterion on speech discrimination than adults. The required intensity level may differ for each child. Therefore, we used multiple intensity levels to assess discrimination. We used the best overall CHT score regardless of intensity level for infants and included all infant’s scores regardless of whether or not they reached criterion.

In contrast, the Tsao et al. [17] study employed the number of trials to reach criterion. Utilizing a total score has important implications for clinical utilization as it reduces the burden on the examiner. The data presented in this study included all children who demonstrated they were conditioned to the task before initiation of the testing protocol. Finally, we presented results at an individual level and included all participants who met the inclusion criteria, regardless of whether their speech discrimination score met criteria.

Furthermore, we examined receptive and expressive spoken language using the MBCDI and the MSEL. In contrast to the relationship between speech discrimination and the MSEL, we did not observe a significant relationship between speech discrimination and the MBCDI expressive language abilities. However, we did observe an association between speech discrimination and the MSEL, thus suggesting that these different measures may be tapping into different cognitive-linguistic skills. Similar to Tsao et al. [17], we examined the relationship between the maternal level of education on language outcomes measured by both the MBCDI and the MSEL. Like Tsao and colleagues, maternal level of education was not a significant predictor of speech discrimination abilities or language abilities measured on the MBCDI. However, Tsao and colleagues examined only CNH. In contrast, maternal level of education was a significant predictor of expressive language measured by the MSEL. Of note, maternal level of education as a predictor variable did not differ between groups and did not predict speech discrimination performance. 

### 4.3. Are the MBCDI and the MSEL Related?

Thirdly, we examined if at 30 months of age, scores from a parent questionnaire of their child’s spoken language inventory significantly correlate with their assessed early receptive and expressive spoken language ability. Assessing MBCDI and MSEL at the same timepoint allowed examination of how a parent’s report of language abilities relates to a clinically administered assessment of receptive and expressive language abilities. The MBCDI was predictive of both receptive and expressive language outcomes measured by the MSEL at 30 months. We employed the MSEL as it allowed direct observation of a child’s ability to follow complex directions and assess their abilities to repeat back complex sentences. However, it does require additional resources to complete, compared to a parent survey. Furthermore, due to the difference in the relationship between infant speech discrimination abilities and each language measure, perhaps differences in these relationships with the MSEL are because it measures beyond lexical or vocabulary skills. Finally, due to the small effect sizes, these results may reflect the need to examine these relationships in a larger population of CHH and CNH and necessitate further examination. 

### 4.4. Comparison of Receptive and Expressive Language

Fourthly, we examined “How receptive and expressive language abilities in CHH compare to CNH?” We were surprised that including receptive and expressive language raw scores as dependent variables and hearing status as the predictor showed no significant effect of hearing status on language outcomes. Our own clinical interests lie in predicting the risks of language difficulties in CHH. Two to three per 1000 children are born yearly with mild to severe hearing loss or deafness [53]. These children go through periods typically characterized by rapid speech and language learning without complete access to the auditory linguistic input due to the nature of hearing loss. Given the critical role of speech discrimination in spoken language learning, CHH may be vulnerable to significant delays or differences in speech discrimination which could be identified during infancy but is not yet standard pediatric audiological follow-through after diagnosis in the infant period. Currently, speech discrimination is not assessed clinically until CHH are at least two years of age [14,15,16]. CHH remain at risk for developing language, regardless of the benefits of early intervention [54,55]. However, no significant differences were observed in our study population for speech discrimination abilities or language outcomes. 

### 4.5. Is There an Effect on Audibility and Hearing Aid Use among CHH

Our final question was the following: among CHH, what effect does audibility and hearing aid use have on early speech discrimination and language abilities?

Aided SII (i.e., audibility) was positively related to CHT performance for /a-i/ and /ba-da/ contrasts when measured at the same timepoint. However, aided SII, neither measured at 9 months (time of CHT) nor measured at 30 months (time of MSEL) significantly correlated with MSEL receptive vocabulary raw scores. These results were unexpected based on recent research findings from the Outcomes in Children with Hearing Loss (OCHL) group, which have reported relationships between aided SII and later language outcomes [18,21,56]. Perhaps these differences in findings are due to little variability (i.e., reported in this manuscript) in aided SII values for children with mild hearing loss (see Figure 6), but greater variance among CHH with ≥moderate hearing losses. Next, we examined the relationship between datalogging (average amount of daily hearing aid use) with speech discrimination and language abilities. Datalogging was not related to discrimination or to MSEL receptive or expressive spoken vocabulary raw scores when measured concurrently (i.e., 9 at 30 months) or when measured at 9 months and examining its relationship with language at 30 months. Again, these results are not in agreement with findings from the OCHL group who examined these relationships among older CHH [18]. 

## 5. Potential Clinical Implications

We have produced a conceptual replication of Tsao et al.’s previous work [17], and our findings support that infant speech discrimination is related to subsequent language abilities not only in CNH but also in CHH. Thus, indicating that the speech discrimination development of CHH who were very early-identified and received early intervention services is not significantly different between groups. This study is the first to show evidence of the similarities in infant speech discrimination abilities between CHH and CNH. 

For a tool of speech discrimination to be clinically useful, speech discrimination must predict not only successful language outcomes, but also outcomes for children at risk for language learning difficulties (i.e., CHH). Speech discrimination ability predicted expressive and receptive language. Early identification of children at risk for developing abnormal language is important. Children identified with expressive language delays [57] and hearing loss [7] who receive early intervention begin school with language similar to their typically developing peers. In contrast, children who receive delayed treatment for expressive language delays [57] and intervention for hearing loss [6] have delayed language throughout elementary school and beyond. 

The results of this study should be viewed in light of certain limitations. First, due to the motivation of clinical utility, participants were not brought back to repeat the assessment of speech discrimination if they could not reach criterion. Additionally, there was little variability among CHH regarding how well their hearing aids were fit; all children benefited from early identification of hearing loss and intervention services. Therefore, these results may not generalize to a different population of CHH. The limitations of this work have helped formulate new lines of inquiry. Future studies will examine the cognitive-linguistic relationship with infant speech discrimination due to the complex nature of language learning and its dynamic processes. 

## 6. Conclusions

These findings are the first to document a relationship between infant speech discrimination measured during the first year of life, using a clinically feasible pre-linguistic speech discrimination assessment and later receptive and expressive language abilities among both CHH and CNH. It is important to note that not all children reached criterion for speech discrimination but including these children in our model still resulted in performance on CHT being a significant predictor of later receptive and expressive language abilities. Establishing the relationship between infant speech discrimination and later language outcomes using a clinically viable protocol is an essential step towards validating the utility of infant speech discrimination as a validation of amplification fitting, and a predictor of later spoken language development.

## Figures and Tables

**Figure 1 jcm-11-05821-f001:**
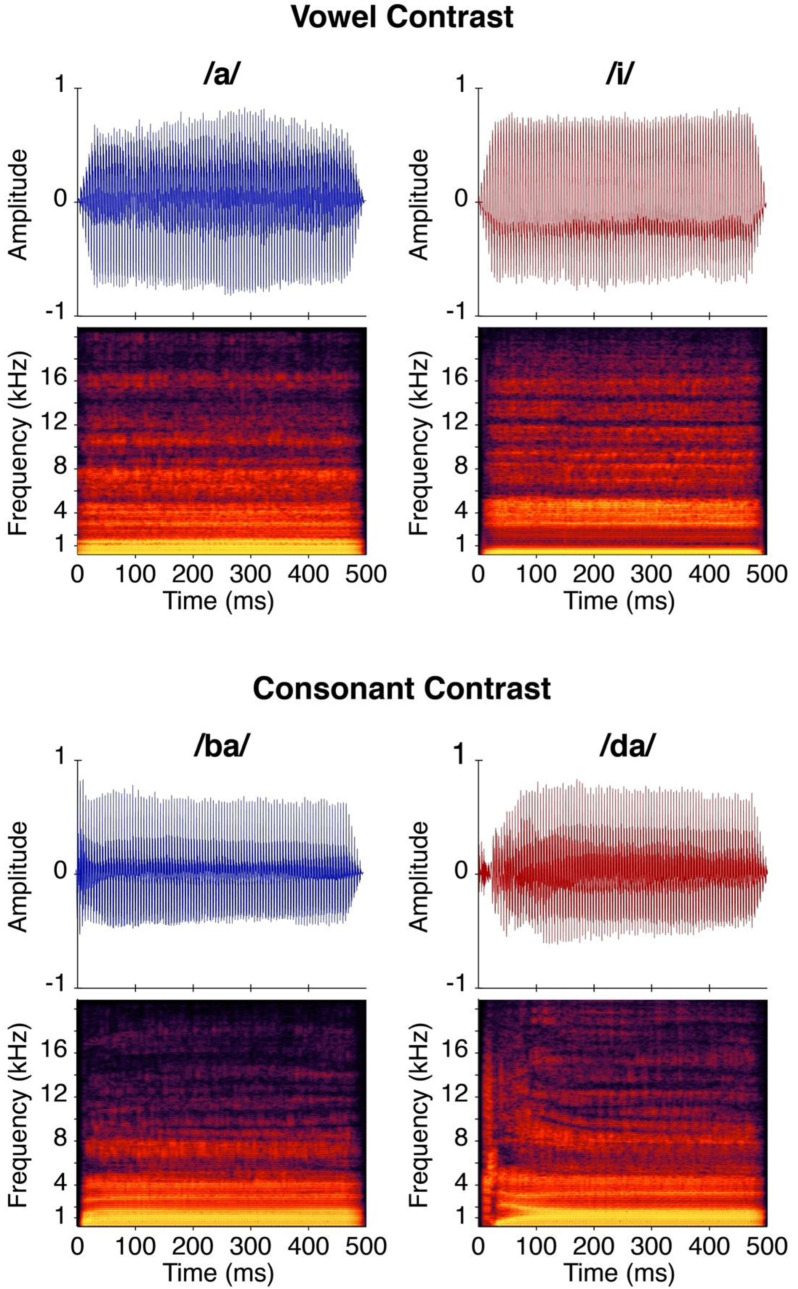
Time-amplitude waveforms and spectrograms for stimuli tested during the speech discrimination task. The top panel shows the /a/ and /i/ speech sounds for the vowel contrast and the bottom panel shows the /ba/ and /da/ speech sounds for the consonant. This figure was previously published in this open access journal, Uhler et al. [12].

**Figure 2 jcm-11-05821-f002:**
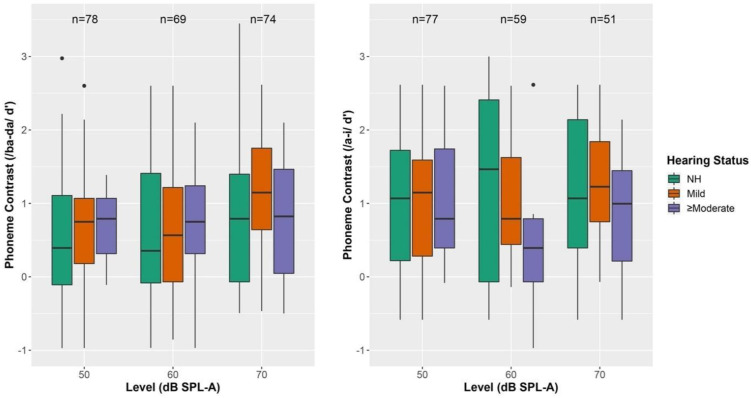
Performance on CHT as a function of presentation level. Note: Normal hearing (NH); Sound Pressure Level A-Weighted (SPL-A) Speech discrimination is shown in d’ for listeners who completed both tasks. The left panel represents performance for the /ba-da/ contrast (y-axis) and the right panel represents performance for the /a-i/ contrast (y-axis). Box plot represents the minimum, first quartile, median, third quartile and maximum. Dots represent participants 2.70 SDs or more from the mean.

**Figure 3 jcm-11-05821-f003:**
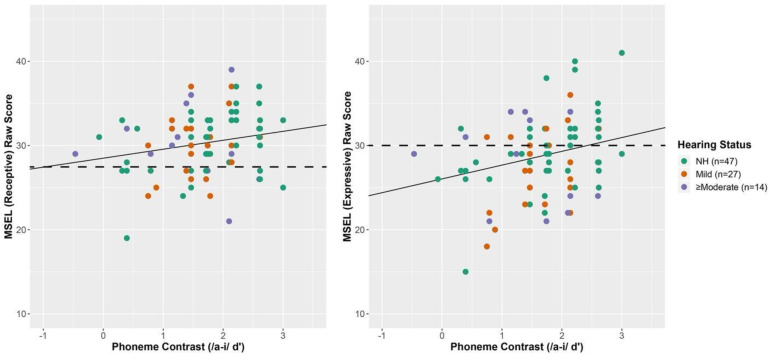
Speech discrimination and maternal level of education as predictors of spoken receptive and expressive language Note. Speech discrimination for the /a-i/ contrast measured in d’ is shown for listeners on the x-axis and receptive (**left panel**) and expressive (**right panel**) spoken language on the y-axis. The regression line was calculated for the mean value of maternal level of education. The dotted line represents the 50th percentile of the age-referenced score. Only 47 of 49 CNH were able.

**Figure 4 jcm-11-05821-f004:**
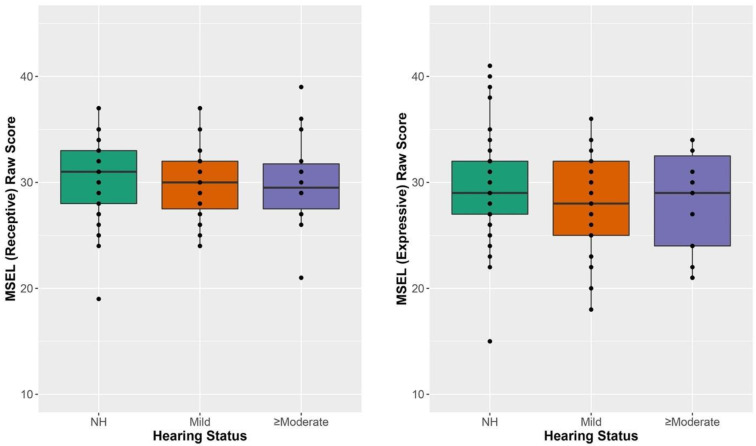
Hearing status as a function of performance on the MSEL Note: Normal Hearing (NH); Mullen Scales of Early Learning (MSEL); Performance on MSEL Receptive (**left panel**) and MSEL Expressive (**right panel**) Hearing status (normal hearing in green, mild in orange, and >moderate in purple). Box plot represents the minimum, first quartile, median, third quartile and maximum. Dots represent participants 2.70 standard deviations or more from the mean.

**Figure 5 jcm-11-05821-f005:**
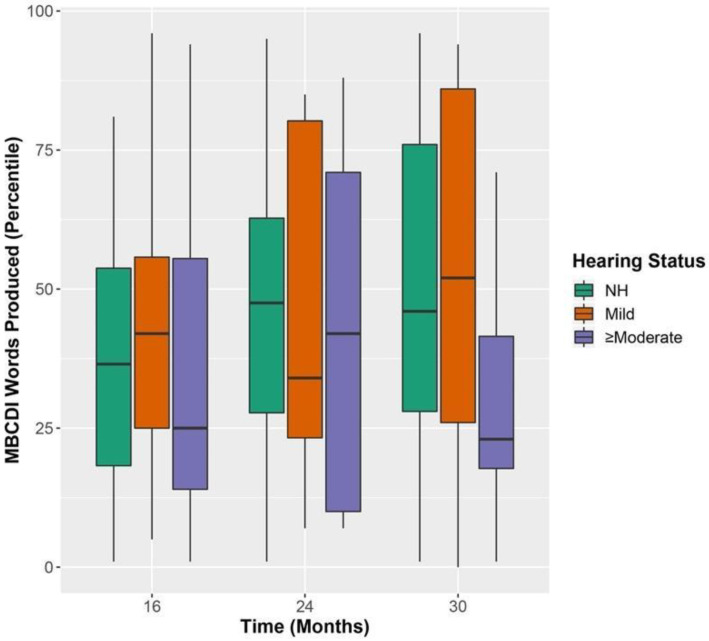
MBCDI performance over time. Note: Normal Hearing (NH); MacArthur-Bates Communicative Development Inventories (MBCDI); MBCDI performance over time (16, 24, and 30 months) as a function of hearing status (normal hearing in green, mild in orange, and ≥moderate in purple). Box plot represents the minimum, first quartile, median, third quartile and maximum. Dots represent participants’ 2.70 standard deviations or more from the mean.

**Figure 6 jcm-11-05821-f006:**
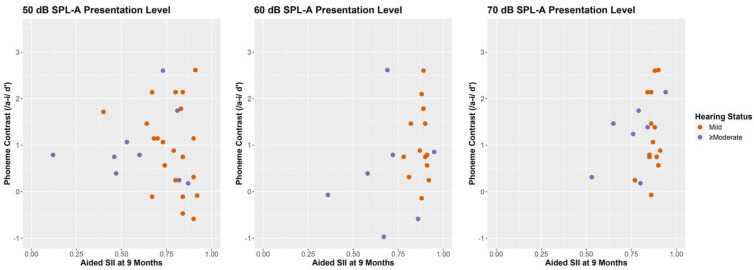
Aided SII at 9 months and speech perception for /a-i/ by presentation level. Note: Speech Intelligibility Index (SII); Sound Pressure Level A-Weighted (SPL-A); Conditioned Head Turn (CHT); Aided speech intelligibility measured at the time of CHT and as a function of CHT discrimination for children with hearing loss (mild in orange, and ≥moderate in purple).

**Table 1 jcm-11-05821-t001:** Overall participant characteristics.

Demographic	CNH	CHH	Statistical Test	*p*-Value
Male	23 (46.9%)	21 (51.2%)		
Age at CHT	M = 8.99 (SD = 1.79)	M = 9.91 (SD = 2.21)	t(88) = 2.18	*p* = 0.03
Age at MSEL	M = 30.61 (SD = 0.71)	M = 30.79 (SD = 0.93)	t(88) = 1.06	*p* = 0.29
Age at MBCDI	M = 30.48 (SD = 0.49)	M = 30.66 (SD = 1.22)	t(82) = 0.89	*p* = 0.38
**Race and Ethnicity**				
White	45 (91.8%)	36 (87.8%)		
Black	2 (4.1%)	0 (0.0%)		
Asian	1 (2.0%)	0 (0.0%)		
More than One Race	1 (2.0%)	5 (12.2%)		
Hispanic/Latino	12 (24.5%)	5 (12.2%)		
**Maternal Level of Education**				
High School Diploma or lower	8 (16.3%)	10 (24.3%)		
Post-secondary attendance	4 (8.2%)	7 (17.1%)		
College Graduate	20 (40.8%)	13 (31.7%)		
*Post-Graduate Degree*	17 (34.7%)	11 (26.9%)		

Note. Age reported in months; Mean (M); Standard deviation (SD) for continuous measures and frequency (percent) for categorical measures. Children who are normal hearing (CNH), Children who are hard of hearing (CHH), Infant speech discrimination (conditioned head turn; CHT), MacArthur-Bates Communicative Development Inventories (MBCDI), Mullen Scale of Early Learning (MSEL).

**Table 2 jcm-11-05821-t002:** Characteristics unique to CHH.

	Mean (SD)	Range (Min, Max)
Age at Hearing Aid Fit (months)	2.79 (0.88)	1.28, 6.17
Datalogging at CHT (average hours/day)	6.39 (4.16)	0.0, 16.0
Missing	4 (9.7%)	
SII at 50 dB at CHT	0.74 (0.19)	0.23, 0.97
SII at 60 dB at CHT	0.83 (0.14)	0.35, 0.96
SII at 70 dB at CHT	0.82 (0.12)	0.44, 0.94
Missing (50 and 70)	2 (4.9%)	
Missing (60)	3 (7.3%)	
Datalogging at MSEL (average hours/day)	8.26 (4.70)	0.10, 19.70
Missing	11(27%)	
SII at 50 dB at MSEL	0.72 (0.18)	0.12, 0.92
SII at 60 dB at MSEL	0.82 (0.13)	0.36, 0.96
SII at 70 dB at MSEL	0.84 (0.09)	0.53, 0.94
Missing at any level	11 (26.8%) *	

Note. Age reported in months, Mean (M); Standard deviation (SD) for continuous measures, and frequency (percent) for categorical measures. Conditioned head turn (CHT), Mullen Scale of Early Learning (MSEL), aided speech intelligibility index (SII), which is unique to hearing aids. * Two children transitioned to cochlear implants between CHT timepoint and MSEL. “Missing” denotes data that was not available.

**Table 3 jcm-11-05821-t003:** Participants who reached criterion in CHT task. *p*-values include Bonferroni corrections for multiple comparisons.

/a-i/ Contrast	CNH	CHH	Chi-Square Test	*p*-Value
60 dB	18 (38%)	7 (17%)	χ^2^(1) = 2.754	0.388
50 dB	14 (30%)	15 (37%)	χ^2^(1) = 0.230	1.000
70 dB	6 (13%)	9 (22%)	χ^2^(1) = 0.922	1.000
Did not reach criterion	9 (19%)	10 (24%)	χ^2^(1) = 0.229	1.000
**/ba-da/ Contrast**				
60 dB	8 (19%)	9 (22%)	χ^2^(1) = 0.127	1.000
50 dB	8 (19%)	5 (12%)	χ^2^(1) = 0.427	1.000
70 dB	4 (14%)	9 (22%)	χ^2^(1) = 1.991	0.633
Did not reach criterion	21 (49%)	17 (42%)	χ^2^(1) = 0.125	1.000

Note. Mean (M); Standard deviation (SD). Number and percentage of participants at the lowest intensity level where criteria (d’ > 1.2) was achieved. If a child reached criteria at a lower level, their score for greater intensity levels is not included. Of note, for /a-i/ 19% of CNH and 24% of CHH and for /ba-da/ 49% of CNH and 42% of CHH did not reach criteria at any level. The final column reports the *p*-values for Chi-Square test, which reveals no significant difference in performance for any contrast at any level.

**Table 4 jcm-11-05821-t004:** Results of multivariate ANOVA predicting sensitivity to /a-i/ and /ba-da/ contrasts.

	**/a-i/ Contrast**					
**Factor**		**F**	**df**	**Error df**	***p*-Value**	**Cohen’s f^2^**
Presentation Level		7.572	2	213	0.0007	0.07
Hearing Status		0.597	2	81	0.553	0.01
	**/ba-da/ Contrast**					
**Factor**		**F**	**df**	**Error df**	***p*-Value**	**Cohen’s f^2^**
Presentation Level		4.225	2	213	0.016	0.04
Hearing Status		0.485	2	81	0.618	0.01

**Table 5 jcm-11-05821-t005:** Results of post-hoc tests on ANOVA in Table 4.

**/a-i/ Contrast**	**t**	**df**	***p*-Value**
50 vs. 60 dB	−2.025	213	0.332
50 vs. 70 dB	−3.890	213	0.002
60 vs. 70 dB	−1.866	213	0.426
**/ba-da/ Contrast**	**t**	**df**	***p*-Value**
50 vs. 60 dB	−2.138	213	0.272
50 vs. 70 dB	−2.775	213	0.066
60 vs. 70 dB	−0.637	213	0.988

**Table 6 jcm-11-05821-t006:** Results of multivariate regression predictive Mullen Scales of Early Learning (MSEL) receptive and expressive spoken vocabulary.

	**Receptive Vocabulary**					
**Factor**		**Estimate (Standard Error)**	**95% Confidence Interval**	**t-Statistic**	***p*-Value**	**Cohen’s f^2^**
Intercept		26.2 (1.38)	[23.5,28.9]	19.03	<0.0001	--
Discrimination of /a-i/ Contrast		1.06 (0.53)	[0.0,2.1]	2.00	0.049	0.07
Maternal Level of Education		0.48 (0.24)	[0.0,1.0]	1.98	0.051	0.05
	**Expressive Vocabulary**					
**Factor**		**Estimate (Standard Error)**	**95% Confidence Interval**	**t-Statistic**	***p*-Value**	**Cohen’s f^2^**
Intercept		21.67 (1.63)	[18.5,24.9]	13.32	<0.0001	--
Discrimination of /a-i/ Contrast		1.64 (0.63)	[0.4,2.9]	2.61	0.011	0.12
Maternal Level of Education		0.93 (0.29)	[0.4,1.5]	3.24	0.002	0.12

Note. Multivariate regression for each model MSEL receptive and expressive language at 30 months as a function of CHT score, for /a-i/ and Maternal Level of Education.

**Table 7 jcm-11-05821-t007:** Results of the Analysis of Variance predicting MacArthur-Bates Communicative Development Inventories (percentile).

	**Receptive Vocabulary**					
**Factor**		**F**	**df**	**Error df**	***p*-Value**	**Cohen’s f^2^**
Time		3.615	2	143	0.029	0.05
Mullen Scales of Early Learning Score		26.885	1	81	<0.0001	0.33
Hearing Status		1.987	2	81	0.144	0.05
Time × MSEL Score		5.121	2	143	0.0071	0.07
	**Expressive Vocabulary**					
**Factor**		**F**	**df**	**Error df**	***p*-Value**	**Cohen’s f^2^**
Time		4.841	2	140	0.009	0.07
MSEL Score		15.038	1	81	0.0002	0.19
Hearing Status		1.601	2	81	0.208	0.04
Time × MSEL Score		7.095	2	140	0.001	0.10

**Table 8 jcm-11-05821-t008:** Results of post-hoc tests on Analysis of Variance presented in Table 7.

**Model Including Receptive Vocabulary**			
**Time**	**t**	**df**	***p*-Value**
16 vs. 24 months	−3.312	141	0.003
16 vs. 30 months	−4.106	143	0.0002
24 vs. 30 months	−0.652	143	0.792
**Model Including Expressive Vocabulary**			
**Time**	**t**	**df**	***p*-Value**
16 vs. 24 months	−3.295	141	0.004
16 vs. 30 months	−4.157	142	0.0002
24 vs. 30 months	−0.718	142	0.753

**Table 9 jcm-11-05821-t009:** Hearing Status, MSEL spoken Receptive and Expressive Raw Scores and MBCDI Percentile at 30 months.

Hearing Status	MSEL Receptive Raw Score Mean (Standard Deviation)	MSEL Expressive Raw Score Mean (Standard Deviation)	MBCDI Percentile at 30 Months Mean (Standard Deviation)
CNH	30.65 (3.65)	29.47 (4.54)	51.59 (29.24)
Mild CHH	29.85 (3.50)	28.00 (4.57)	50.92 (32.66)
≥Moderate CHHs	30.14 (4.52)	28.07 (4.92)	31.50 (23.01)

Note. CNH = Children with normal hearing, MSEL = Mullen Scales of Early Learning, MBCDI = MacArthur Bates Communicative Development Inventory. Hearing status: Normal hearing 0–15 dB HL, Mild hearing loss = 16–39 dB HL, Moderate hearing loss ≥40 dB HL.

**Table 10 jcm-11-05821-t010:** Correlations between aided Speech Intelligibility Index (SII), speech contrast, datalogging measures, and Mullen Scales of Early Learning (MSEL) receptive and expressive vocabulary scores.

Variables	Pearson r	df	*p*-Value
Aided SII (50 dB) vs. /a-i/ discrimination	−0.016	36	0.923
Aided SII (60 dB) vs. /a-i/ discrimination	0.334	21	0.120
Aided SII (70 dB) vs. /a-i/ discrimination	0.513	23	0.009
Aided SII (50 dB) vs. /ba-da/ discrimination	0.153	32	0.365
Aided SII (60 dB) vs. /ba-da/ discrimination	0.065	35	0.716
Aided SII (70 dB) vs. /ba-da/ discrimination	0.278	32	0.096
Aided SII (50 dB at 9 mos) vs. MSEL receptive vocabulary	−0.097	37	0.556
Aided SII (60 dB at 9 mos) vs. MSEL receptive vocabulary	−0.114	36	0.496
Aided SII (70 dB at 9 mos) vs. MSEL receptive vocabulary	−0.094	37	0.569
Aided SII (50 dB at 9 mos) vs. MSEL expressive vocabulary	−0.091	37	0.582
Aided SII (60 dB at 9 mos) vs. MSEL expressive vocabulary	−0.057	36	0.732
Aided SII (70 dB at 9 mos) vs. MSEL expressive vocabulary	−0.094	37	0.551
Aided SII (50 dB at 30 mos) vs. MSEL receptive vocabulary	−0.090	28	0.638
Aided SII (60 dB at 30 mos) vs. MSEL receptive vocabulary	−0.070	28	0.711
Aided SII (70 dB at 30 mos) vs. MSEL receptive vocabulary	−0.048	28	0.799
Aided SII (50 dB at 30 mos) vs. MSEL expressive vocabulary	0.030	28	0.876
Aided SII (60 dB at 30 mos) vs. MSEL expressive vocabulary	0.086	28	0.652
Aided SII (70 dB at 30 mos) vs. MSEL expressive vocabulary	0.115	28	0.545
Datalogging (9 mos) vs. MSEL receptive vocabulary	−0.259	35	0.122
Datalogging (9 mos) vs. MSEL expressive vocabulary	−0.240	35	0.153
Datalogging (9 mos) vs. maternal level of education	0.074	35	0.662
Datalogging (30 mos) vs. MSEL receptive vocabulary	−0.072	28	0.704
Datalogging (30 mos) vs. MSEL expressive vocabulary	0.020	28	0.916
Datalogging (30 mos) vs. maternal level of education	0.466	28	0.009

## Data Availability

De-identified data will be made available after institutional review and approval.

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
