# Peer review of "Speech Discrimination in Infancy Predicts Language Outcomes at 30 Months for Both Children with Normal Hearing and Those with Hearing Differences"

_jcm, 2022, doi:10.3390/jcm11195821_

Round 1
Reviewer 1 Report
Dear Authors,
I read you interesting work entitled “Speech Discrimination in Infancy Predicts Language Outcomes at 30 months for both Children with Normal Hearing and those with Hearing Differences” and here I enclose my recommendations:
General Comments
1. Citations: Your manuscript does not follow proper numbering of your references and that causes several issues in text in all sections and subsections. Please correct them (I have some examples from your Introduction)
“….. Since the inception of UNHS/EHDI, research has revealed a wide array of demographic characteristics, such as a lesser degree of hearing loss [19] and earlier age at amplification [20], which are related to improved language outcomes and cognitive abilities [17–19]. Ching et al [20,21] reported that maternal level of education, degree of hearing loss, gender, and presence of additional disabilities were also major predictors of language outcomes. While understanding the relationships between these characteristics and language outcomes is important, variables such as gender, maternal level of education, and degree of hearing loss cannot be altered to impact outcomes. However, performance on speech discrimination can be used to guide interventions. For example, the assessment of speech discrimination in older children [26] and adults [23,24] with hearing differences validates that the amplification fitting, assuring that individuals with hearing differences have access to the sounds of their spoken language, is related to communication. Performance on speech discrimination guides interventions such as hearing aid programming…..”
2. English: Your manuscript is informative and included lots of data especially in the “Introduction” and the “Methods” section. This is good but on the other hand it is not readers friendly since several syntax errors occurs. I suggest the Authors to have the text for a proofread and limit it to basic information. This will make the manuscript more solid and more “to the point”.
Introduction
There are many parts in this section that has no citation, or you mention that many studies referred to something important. I suggest the Authors to address this. Here are some examples from your manuscript:
“… These studies demonstrated the feasibility of both EEG/MMR and CHT/VRISD procedures to assess infant speech discrimination in the first year of life…. “
“….. Without speech discrimination assessments that can be administered in the first three years of life to children with hearing differences, validation of the amplification fitting cannot be done until 3 years have passed. Modifications and development of early intervention strategies will be limited for listening and spoken language without additional information about a child’s prelinguistic speech discrimination skills. No studies investigating the relationship between speech discrimination measured during infancy and language outcomes among CHH who have benefited from UNHS/EHDI programs were found in the literature. Until recently, assessment of speech discrimination during infancy and toddlerhood has yielded only group results rather than individual scores. This prevented examining the relationship of an individual child’s speech discrimination to later language development….. “
“…. 4.1 Speech Discrimination
The majority of infants discriminated /a-i/ at one of three intensity levels (50-, 60-, or 563 70-dB SPL-A). There was not a significant difference in speech discrimination abilities between CHH and CNH. Discrimination of the /ba-da/ contrast was observed in fewer infants across both groups. Of note, all CHH were identified through newborn hearing screenings, were fit with hearing aids and enrolled in early intervention by 6 months of age. Among CHH, there was no difference in speech discrimination abilities as a function of degree of hearing loss (i.e., mild versus ≥moderate). Though non-significant findings are generally not discussed, our findings that there were no significant differences between CHH and CNH groups for discrimination of either /a-i/ or /ba-da/, and that within the CHH group there was no significant difference by degree of hearing loss are theoretically important. They highlight the effectiveness of the early identification/intervention as a result of UNHS/EHDI for these children with hearing differences on speech discrimination abilities in infancy, abilities that would be most negatively affected without early identification/intervention and could set off a cascade of delayed development in spoken language for these children.
While all children demonstrated conditioning to the task before testing began, bringing children back may have resulted in children being able to discriminate the /ba-da/ contrast. Because our aim was to examine the clinical utility of this CHT procedure we did not design the study for return visits. The logistics of having parents return within a short timeframe would likely increase subject attrition and be challenging to accommodate in future clinical settings. Additionally, assessing speech discrimination for one contrast took an average of 5 minutes which is feasible to complete in a clinical audiology setting among young infants…...”
Results
It is very interesting that the Authors formed their results in accordance with their research questions, but this suits better to the discussion section. I suggest the Authors to use subtitles and subsections as the Journal Suggests.
Discussion
The section 4.1. in not that clear presented without literature support.
In section 4.2. there is no clear evidence that vowel discrimination relates to language development. I suggest the reviewers to clarify more this.
I suggest the Authors to consider the above, because with this they will support better their clinical implication and their conclusion.
Thank you.
Author Response
Dear Reviewer.
Thank you for taking time to review our manuscript. We feel your suggestions have been very helpful in improving this paper.
Below are point-by-point responses.
General Comments
Point 1: 1. Citations: Your manuscript does not follow proper numbering of your references and that causes several issues in text in all sections and subsections. Please correct them (I have some examples from your Introduction)
“….. Since the inception of UNHS/EHDI, research has revealed a wide array of demographic characteristics, such as a lesser degree of hearing loss [19] and earlier age at amplification [20], which are related to improved language outcomes and cognitive abilities [17–19]. Ching et al [20,21] reported that maternal level of education, degree of hearing loss, gender, and presence of additional disabilities were also major predictors of language outcomes. While understanding the relationships between these characteristics and language outcomes is important, variables such as gender, maternal level of education, and degree of hearing loss cannot be altered to impact outcomes. However, performance on speech discrimination can be used to guide interventions. For example, the assessment of speech discrimination in older children [26] and adults [23,24] with hearing differences validates that the amplification fitting, assuring that individuals with hearing differences have access to the sounds of their spoken language, is related to communication. Performance on speech discrimination guides interventions such as hearing aid programming…..”
Response 1: We have addressed these concerns and made sure the numbering is sequential as required by this citation type. These corrections are throughout the document.
Point 2. English: Your manuscript is informative and included lots of data especially in the “Introduction” and the “Methods” section. This is good but on the other hand it is not readers friendly since several syntax errors occurs. I suggest the Authors to have the text for a proofread and limit it to basic information. This will make the manuscript more solid and more “to the point”.
Response 2: We have removed some text and attempted to ensure clarity for the reader.
Introduction
Point 3: There are many parts in this section that has no citation, or you mention that many studies referred to something important. I suggest the Authors to address this. Here are some examples from your manuscript:
“… These studies demonstrated the feasibility of both EEG/MMR and CHT/VRISD procedures to assess infant speech discrimination in the first year of life…. “
Response 3: We have deleted this paragraph as it seemed redundant with later text within the introduction.
Point 4 “….. Without speech discrimination assessments that can be administered in the first three years of life to children with hearing differences, validation of the amplification fitting cannot be done until 3 years have passed. Modifications and development of early intervention strategies will be limited for listening and spoken language without additional information about a child’s prelinguistic speech discrimination skills. No studies investigating the relationship between speech discrimination measured during infancy and language outcomes among CHH who have benefited from UNHS/EHDI programs were found in the literature. Until recently, assessment of speech discrimination during infancy and toddlerhood has yielded only group results rather than individual scores. This prevented examining the relationship of an individual child’s speech discrimination to later language development….. “
Response 4: We completed an additional literature search and did not find any manuscripts concerning the relationship between infant speech perception and language outcomes among infants with hearing loss for the ages reported in this paper. We would be happy to provide any additional, specific references upon request.
Point 5 “…. 4.1 Speech Discrimination
The majority of infants discriminated /a-i/ at one of three intensity levels (50-, 60-, or 563 70-dB SPL-A). There was not a significant difference in speech discrimination abilities between CHH and CNH. Discrimination of the /ba-da/ contrast was observed in fewer infants across both groups. Of note, all CHH were identified through newborn hearing screenings, were fit with hearing aids and enrolled in early intervention by 6 months of age. Among CHH, there was no difference in speech discrimination abilities as a function of degree of hearing loss (i.e., mild versus ≥moderate). Though non-significant findings are generally not discussed, our findings that there were no significant differences between CHH and CNH groups for discrimination of either /a-i/ or /ba-da/, and that within the CHH group there was no significant difference by degree of hearing loss are theoretically important. They highlight the effectiveness of the early identification/intervention as a result of UNHS/EHDI for these children with hearing differences on speech discrimination abilities in infancy, abilities that would be most negatively affected without early identification/intervention and could set off a cascade of delayed development in spoken language for these children.
While all children demonstrated conditioning to the task before testing began, bringing children back may have resulted in children being able to discriminate the /ba-da/ contrast. Because our aim was to examine the clinical utility of this CHT procedure we did not design the study for return visits. The logistics of having parents return within a short timeframe would likely increase subject attrition and be challenging to accommodate in future clinical settings. Additionally, assessing speech discrimination for one contrast took an average of 5 minutes which is feasible to complete in a clinical audiology setting among young infants…...”
Response 5: This is a summary from our findings reported in this paper, therefore we did not reference any other studies at this point. We have tried to clarify this on page 19 by adding this “Our results reported here, revealed that the majority of infants discriminated /a-i/ at one of three intensity levels (50-, 60-, or 70-dB SPL-A).”
Additionally, in lines 649-654 we expand on our statement about the cascade of delayed development.
Results
Point 6: It is very interesting that the Authors formed their results in accordance with their research questions, but this suits better to the discussion section. I suggest the Authors to use subtitles and subsections as the Journal Suggests.
Response 6: We respectfully disagree, the subtitles within the results section are not specified (https://www.mdpi.com/journal/jcm/instructions#manuscript).
Discussion
Point 7: The section 4.1. in not that clear presented without literature support.
Response 7: We have clarified this, please see above.
Point 8: In section 4.2. there is no clear evidence that vowel discrimination relates to language development. I suggest the reviewers to clarify more this.
Response 8: We have added additional references demonstrating the relationship between infant speech perception and later language abilities among normal hearing children. We have added a statement about the /ba-da/ contrast: We were surprised that the same relationship was not observed for the /ba-da/ contrast. Perhaps it is due to overall poorer performance on /ba-da/ or lack of variability in performance on this contrast. However, our study design did not allow us to examine “why” /ba-da/ was not related to later language development.
Furthermore, due to the reviewer’s concern of discussion, we have made marked changed throughout the discussion. We are grateful for these suggestions, as we feel it has substantially improved the manuscript.

Reviewer 2 Report
Speech Discrimination in Infancy Predicts Language Outcomes 2 at 30 months for both Children with Normal Hearing and those 3 with Hearing Differences
Introduction:
Very comprehensive and well written, however, it suffers some ambiguous and long statements. Some of them were underlined
The rationale of the work: is very clear
Materials and Methods:
· Participants:
Lines 140-144: their description is somewhat ambiguous with abbreviations. Rewrite in a more comprehensive way.
Lines 148-153: Is the non-included data belong to this study or the previous one?
Lines 180-181: “The mean data logging recorded was 180 6.39 hours (SD=4.16) for CHT and increased at MSEL to 8.26 averaged hours per day (4.70)”. Why there is a difference in data logging between CHT and MSEL?? Explain, please.
· Stimuli:
Lines 228-229: The presentation level of stimuli (either 50, 60, and/or 70 dB SPL-A) is dependent on what???
The authors did not mention whether the target stimulus was presented randomly or not.
Results:
There is no need to repeat the f, r2 P, and other values of the statistics in the text portion of the results. Too confusing and the reader can refer to tables. Ex: lines 517- 534???
Speech Discrimination:
Line 292: what the authors mean by “regardless of performance at 70 dB SPL-A”
Discussion:
Well written but avoid long statement
References:
Reference 14 is missing the journal title.
Tables:
Table (1): the authors should refer that age was measured in months
Table (2): What do you mean by the word “Missing”?

Author Response
Dear Reviewer 2. We are grateful for your time, edits and suggestions. These suggestions have significantly improved this manuscript. Below are point-by-point responses. I have also attached a revised manuscript with these changes tracked, for ease of your review.
Point 1:
Introduction:
Very comprehensive and well written, however, it suffers some ambiguous and long statements. Some of them were underlined
Response 1: We have removed some text and attempted to ensure clarity for the reader. We have denoted deleted text using the strikethrough of text and changes are in red text throughout the revised document.
Point 2: Lines 140-144: their description is somewhat ambiguous with abbreviations. Rewrite in a more comprehensive way.
Response 2: We have re-written to removed stats which have been placed into tables and consolidated text for readability (lines: 141-149)
Our participant section has been reduced and now reads “Data from 90 infants participating in an ongoing longitudinal study were analyzed. All participants’ hearing was screened via universal newborn screening, and results per parent report were recorded. Participants included 41 CHH (21(M), 20(F)) and 49 CNH (23(M), 26(F)). All were born full-term and healthy (see Table 1). CHH infants were approximately one month older at CHT testing, but other assessments occurred at similar ages. All CHH were enrolled in early intervention and fit with hearing aids by six months of age (see Table 2).”
Point 3: Lines 148-153: Is the non-included data belong to this study or the previous one?
Response 3: We have clarified that this is for the current study. “The data for the CHT results for 40 CHH and CNH were previously reported [14]. Data from an additional 18 CHH and 19 CNH were not included in the current study because they did not have both CHT and MSEL data, for the following reasons: COVID shut downs (6 CHH, 3 CNH), diagnosed with secondary disability (1 CHH), lost to follow up (4 CHH, 12 CNH), primary spoken language in the home was not English (2 CHH), family relocated (4 CHH, 3 CNH), different testing protocol was used (1 CNH), could not complete conditioned head turn (1 CHH).”
Point 4: Lines 180-181: “The mean data logging recorded was 180 6.39 hours (SD=4.16) for CHT and increased at MSEL to 8.26 averaged hours per day (4.70)”. Why there is a difference in data logging between CHT and MSEL?? Explain, please.
Response 4: Data collection for the CHT and MSEL occurred at different time points, so the data logging associated with each measurement also occurred at two different time points. We have changed this statement to on page 5 (lines 185-188) to read “While hearing aid usage increased an average of two hours per day between CHT testing (nine months) and MSEL assessment (30 months), the increase was not significant, paired t-test (t(28)=-1.827, p=0.079).”
Additionally, this is one of two examples where we kept the stats within the body of the document rather than moving it to a table.
Point 5: Lines 228-229: The presentation level of stimuli (either 50, 60, and/or 70 dB SPL-A) is dependent on what???
Response 5: We have added the following text as stated here: Once 15 trials at 70 dB SPL-A were completed, the presentation level was reduced to 60 dB SPL-A, and 15 trials were completed at that presentation level, regardless of if the child reached criterion or not, at 70 dB SPL-A.
Point 6: The authors did not mention whether the target stimulus was presented randomly or not.
Response 6: On page 7, lines 272-273 States: Computer software determined trial-type presentation, with either 7 or 8 of the 15 trials being a change or no-change trial as randomly determined by the computer; the evaluator was blind to trial type.
Results:
Point 7: There is no need to repeat the f, r2 P, and other values of the statistics in the text portion of the results. Too confusing and the reader can refer to tables. Ex: lines 517- 534???
Response 7: Thank you for these suggestions, this has significantly improved the readability of the text. We have removed all statistical test to tables (we have added 3 new tables) except for two places within the body of the text where there was no other logical place for the stats to be reported.
Speech Discrimination:
Point 8: Line 292: what the authors mean by “regardless of performance at 70 dB SPL-A”
Response 8: We have clarified this in lines 274- Once 15 trials at 70 dB SPL-A were completed, the presentation level was reduced to 60 dB SPL-A, and 15 trials were completed at that presentation level, regardless of whether or not a child reached criterion, at 70 dB SPL-A.
Discussion:
Point 9: Well written but avoid long statement
Response 9: We have reduced long statements as much as possible, all modified text has been clearly marked in the version of the paper titled track changes.
References:
Point 10: Reference 14 is missing the journal title.
Response 10: We have added in the journal on line 751 K. M. Uhler, A. M. Kaizer, K. A. Walker, and P. M. Gilley, “Relationship between Behavioral Infant Speech Perception and Hearing Age for Children with Hearing Loss,” J Clin Med. vol. 10, no. 19, pp. 1–17, 2021.
Tables:
Point 11: Table (1): the authors should refer that age was measured in months
Response 11: We have added in months to the Note. “Age reported in months;”
Point 12: Table (2): What do you mean by the word “Missing”?
Response12: We have added this statement into the note for table 1“Missing” denotes data that was not available.

Round 2
Reviewer 1 Report
Dear Authors,
I read your work and i have no further comments to do on your study. The only minor point is to have your text again reviewed for typing errors as well as a final editing in English.
Thank you.
Author Response
Reviewer Point:
The only minor point is to have your text again reviewed for typing errors as well as a final editing in English.
Author Response: We have re-read the manuscript and reviewed the manuscript for errors. We identified a few minor errors and inconsistent uses of acronyms. We have made corrections and ensured consistent usage of acronyms.
